# Advances and Prospects of Fowl Adenoviruses Vaccine Technologies in the Past Decade

**DOI:** 10.3390/ijms26136434

**Published:** 2025-07-04

**Authors:** Chunhua Zhu, Pei Yang, Jiayu Zhou, Xiaodong Liu, Yu Huang, Chunhe Wan

**Affiliations:** 1Fujian Provincial Key Laboratory for Avian Diseases Control and Prevention, Institute of Animal Husbandry and Veterinary Medicine, Fujian Academy of Agricultural Sciences, Fuzhou 350013, China; zchlxd80@163.com (C.Z.);; 2Biotechnology Institute, Fujian Academy of Agricultural Sciences, Fuzhou 350013, China

**Keywords:** fowl adenovirus, advances, serotypes, vaccine design, prospects

## Abstract

Over the past decade, diseases associated with fowl adenoviruses (FAdVs) have exhibited a new epidemic trend worldwide. The presence of numerous FAdVs serotypes, combined with the virus’s broad host range, positions it as a significant pathogen in the poultry industry. In the current context of intensive poultry production and global trade, co-infections involving multiple FAdVs serotypes, as well as co-infections with FAdVs alongside infectious bursal disease or infectious anemia virus, may occur within the same region or even on the same farm. The frequency of these outbreaks complicates the prevention and control of FAdVs. Therefore, the development of effective, targeted vaccines is essential for providing technical support in the management of FAdVs epidemics. Ongoing vaccine research aims to improve vaccine efficacy and address the challenges posed by emerging FAdVs outbreaks. This review focuses on vaccines developed and studied worldwide for various serotypes of FAdVs in the past decade. It encompasses inactivated vaccines, live attenuated vaccines, e.g., host-adapted attenuated vaccines and gene deletion vaccines, viral vector vaccines, and subunit vaccines (including VLP proteins and chimeric proteins). The current limitations and future development directions of FAdVs vaccine development are also proposed to provide a reference for new-generation vaccines and innovative vaccination strategies against FAdVs, as well as for the rapid development of highly effective vaccines.

## 1. Introduction

Fowl adenoviruses (FAdVs) are classified within the *Adenoviridae* family, the *Aviadenovirus* genus. According to the guidelines established by the International Committee on Taxonomy of Viruses (ICTV), FAdVs are further categorized into five species: *Aviadenovirus ventriculi* (FAdV-1), *Aviadenovirus gallinae* (FAdV-2, FAdV-3, FAdV-9 and FAdV-11), *Aviadenovirus hydropericardii* (FAdV-4, FAdV-10), *Aviadenovirus quintum* (FAdV-5), and *Aviadenovirus hepatitidis* (FAdV-6, FAdV-7, FAdV-8a and FAdV-8b) [1]. This classification is crucial for understanding the diversity and pathogenic potential of FAdVs serotypes [2,3]. Adenoviruses are non-enveloped viruses with double-stranded DNA. The virus exhibits an icosahedral shape and has a particle size of approximately 70 nm. The virus particles are arranged in a crystal lattice pattern in the nucleus post-infection (Figure 1). FAdVs contain approximately 43 to 46 kilobases of double-stranded genomic DNA, encoding a variety of structural and non-structural proteins. The viral capsid is composed of Hexon, Penton base, and Fiber [4,5]. These components play crucial roles in the stability, virulence, and infection efficiency of the virus [6,7]. Hexon is the most abundant protein component of adenovirus. The viral capsid structure is primarily composed of Hexon, which exists in the form of trimers. Hexon is associated with the type of antibody produced, the specific antigenic determinants of the major genus, and the minor species-specific antigenic determinants. The highly variable region of Hexon is significant for distinguishing different FAdV serotypes. These regions are closely linked to antigenic variation and immune evasion, are primarily responsible for antibody binding, and are frequently utilized to analyze the genetic evolutionary relationships of FAdVs [8,9]. The Penton is located at each vertex of the icosahedral structure and consists of pentameric subunits anchored to the Fibers, which are closely related to viral endocytosis. The Fiber protrudes from the capsid in a ball-and-stick configuration and is essential for the initial attachment of the virus to host cells. Each Fiber comprises a central axis and a knob domain that is involved in receptor recognition. The knob domain interacts with the specific cell surface receptor CAR, facilitating virus entry into host cells [10,11]. In addition to the major capsid proteins, FAdVs also possess minor coat proteins, including IIIa, VI, and IX, which contribute to capsid stability and assembly. These minor proteins play critical roles in virus maturation and disassembly during the infection process [12]. Studying the structural proteins of FAdVs is critical for developing effective vaccines and therapeutic strategies.

Clinical cases of FAdVs infection have been extensively documented in poultry populations worldwide, with an increase in endemic areas resulting in significant economic losses to the poultry industry. FAdVs infect a diverse range of host species, including chickens [13,14], ducks [15,16,17], geese [18], peacocks [19], falcons [20], Mandarin ducks [21], and Black-necked Cranes [22]. Furthermore, FAdVs could cross host species barriers, infecting both poultry and wild birds, which raises concerns regarding the potential for the transboundary emergence of new pathogenic strains [23,24]. The transmission of FAdVs occurs horizontally via contaminated feed, drinking water, excrement, or direct contact in broiler and breeder flocks, as well as vertically through the yolk sac, chorioallantoic membrane, and allantoic cavity of chick embryos in breeder flocks [25,26]. The role of FAdVs vaccines in vertical transmission is primarily manifested in their ability to prevent breeders from transmitting pathogens to embryos and chicks, thereby reducing the disease incidence.

FAdVs infection can lead to various clinical syndromes. Notably, FAdV-1 is associated with adenoviral gizzard erosion, whereas FAdV-4 is linked to hepatitis-hydropericardium syndrome (HHS) [27]. Additionally, FAdV-2, FAdV-11, FAdV-8a, and FAdV-8b are known to cause inclusion body hepatitis (IBH) [28,29]. The incidence of domestic FAdV infection has gradually increased in recent years, with outbreaks of HHS disease caused by a novel strain of FAdV-4 emerging in most provinces of China since 2015 [13,30]. This strain spreads rapidly and has a mortality rate as high as 80% [31], posing a significant threat to the poultry industry. Furthermore, co-infections with multiple serotypes, as well as mixed infections with other viruses, exacerbate the detrimental impact of the disease [32,33].

Various strains of FAdVs have been isolated from deceased or diseased poultry, with numerous serotypes exhibiting significant differences in pathogenicity. Research on FAdVs has primarily focused on elucidating their pathogenicity and early immune responses [34,35]. Researchers worldwide have developed and assessed various vaccines aimed at controlling HHS in poultry flocks. Large-scale vaccination efforts have significantly reduced the incidence of FAdV infections in China. However, multiple FAdVs serotypes continue to circulate within the same region or same chicken farm [32]. While immune protection for a specific serotype is effective, it provides only limited or no protection against other serotype viruses [36]. An analysis of 27 novel FAdVs genomes by Michael Hess’s team at the University of Veterinary Medicine in Vienna, Austria revealed that natural recombination predominantly occurs in two species, FAdV-D and FAdV-E, with FAdV-E exhibiting an unusually high rate of recombination [36,37]. Although chickens vaccinated with the Fiber vaccine develop immunity to homotypic FAdV-8a, they remain susceptible to heterotypic FAdV-8b infections [36]. Furthermore, research has confirmed that co-infection of FAdVs with immunosuppressive diseases, such as infectious bursal disease or infectious anemia virus, can increase the pathogenicity of FAdVs and compromise herd immunity, thereby facilitating the invasion of other pathogens [9,33]. Consequently, in the context of intensive poultry production and global trade, the prevention and control of FAdVs diseases continue to pose significant challenges.

Currently, vaccine development has emerged as a significant area of research for FAdVs. With the ongoing advancements and innovations in molecular biology technologies within the field of vaccinology, investigators have developed various vaccines to control FAdVs. This article reviews inactivated vaccines [38], live attenuated vaccines (including host-adapted attenuated vaccines, gene deletion vaccines) [39,40], viral vector vaccines, and subunit vaccines (including VLP proteins and chimeric proteins) of FAdVs [41,42]. Additionally, it provides an outlook on the future directions of vaccine development, serving as a reference for the creation of more efficient and safer vaccines to manage FAdVs epidemics (Figure 2).

## 2. Inactivated Vaccines

Inactivated vaccines can be categorized into tissue homogenate inactivated vaccines and whole virus inactivated vaccines. The former is typically derived from the liver and other tissues of chickens infected with avian viral diseases, which are processed into vaccines following inactivation. In contrast, the latter is produced by culturing and inactivating purified virus strains. Traditional inactivated vaccine immunization remains the primary method for preventing certain poultry diseases, such as *Mycoplasma gallisepticum* and salmonellosis [43], and this type of vaccine can be rapidly deployed for clinical use. Whole-virus inactivated vaccines, however, have a longer development cycle and require culture in primary or passaged cells for stable viral propagation. For some poultry pathogens, inactivated vaccine immunization is the principal preventive strategy, offering advantages such as high safety, minimal interference from maternal antibodies, and effective humoral immunity.

Several regions of China have experienced novel FAdV-4 epidemics this decade [13,30,44]; inactivated vaccine immunization is the primary method for preventing and controlling FAdV-4, effectively enhancing the host immune response and reducing the incidence of FAdV infection and associated economic losses. Pan et al. isolated and characterized a highly virulent FAdV-4 HLJFAd15 strain from infected laying hens [14]. They utilized an inactivated oil emulsion vaccine derived from this virulent strain to immunize SPF chickens and evaluate both the immune response and protective efficacy [45]. The results demonstrated that the vaccine can induce high levels of specific antibodies and elicit a robust Th2 immune response, providing 100% protection against lethal dose challenge with FAdV-4. Similarly, inactivated vaccines developed from new genotypes of FAdV-4, including SDJN0105 [46] and CH/GZXF/1602 [47], generated high antibody levels and exhibited significant protection in SPF chickens. While numerous studies have focused on the immune protection offered by inactivated vaccines against a challenge with the same FAdV-4 serotype, some research has indicated that the FAdV-4 inactivated vaccine also provides cross-protection against other FAdV serotypes, such as FAdV-8a and FAdV-11. Furthermore, the administration of the FAdV-4 inactivated vaccine has been shown to protect poultry offspring from infections caused by FAdVs of different serotypes [38,47].

Recent advancements have been made in the research of viral vector-based inactivated vaccines. Lu et al. [48] developed a recombinant FAdV-4 strain (FA4-F8b) that expresses FAdV-8b Fiber via CRISPR-Cas9 and homologous recombination technology. The challenge assay demonstrated that inactivated FA4-F8b not only elicited high levels of neutralizing antibodies in chickens but also provided effective protection to 2-week-old chickens against challenges from both the FAdV-4 and FAdV-8b strains. In the same year, Wang et al. [49] constructed a FAdV-4 chimeric virus rFAdV-4-fiber/8b, containing Fiber of FAdV-8b. This virus demonstrated comparable in vitro replication capability and in vivo pathogenicity to the parental wild-type FAdV-4. The inactivated rFAdV-4-fiber/8b vaccine has been shown to confer complete protection against both FAdV-4 and FAdV-8b infections in SPF chickens.

Currently, inactivated vaccines for FAdV-4 have been commercialized. However, the production cost of high-titer inactivated FAdV-4 vaccines is relatively high because of the necessity of culturing a substantial number of viruses to obtain sufficient antigens. Furthermore, the preparation of inactivated vaccines often faces challenges such as inadequate virus titers (viral load) and improper inactivation of vaccine strains. As a result, the immune protection effect of the vaccine or the serum antibody titer may not reach the desired levels. Additionally, the limitations of inactivated vaccines include their inability to elicit robust cellular immunity and the lack of long-lasting immunity [50]. A summary of inactivated vaccines against FAdVs is presented in Table 1.

## 3. Live Attenuated Vaccines

Currently, the production of live attenuated vaccines is predominantly achieved through three main approaches: animal adaptation to a new host or the attenuation of strains via continuous cell passage, screening for naturally attenuated strains, and the deletion of virulence-related genes via molecular biology techniques such as CRISPR-Cas and reverse genetics. These vaccines offer several advantages, such as the use of a minimal amount of antigen, strong immunogenicity, and low cost. A key principle in the design of attenuated vaccines is to ensure the immunogenicity of the strain while simultaneously reducing its virulence [52].

The traditional method of attenuation through continuous cell passage has been employed in the development of FAdVs vaccines (Table 2). Schonewille et al. [39] successfully obtained the attenuated strain FAdV-4/QT35 by culturing virulent FAdV-4 strains in fibroblast cell lines. Notably, no adverse clinical symptoms or mortality were observed following the inoculation of 1-day-old poultry, suggesting that FAdV-4/QT35 is a promising candidate strain for the development of attenuated host-adapted vaccines.

With the advancement of molecular biology technologies, CRISPR/Cas9 genome editing has emerged as a powerful tool for analyzing virus–host interactions and developing new poultry vaccines [53]. Through the genetic modification of viral genomes, researchers have developed several viral vector-based vaccines with improved efficacy [53]. Xie et al. generated the recombinant virus FA4-EGFP, which expresses the EGFP-Fiber-2 fusion protein through CRISPR/Cas9 technology [40]. Compared with the inactivated vaccine, inoculation of FA4-EGFP in chickens not only induced earlier production of neutralizing antibodies but also resulted in higher antibody titers, effectively resisting the fatal challenge posed by FAdV-4, highlighting the potential application value of the FA4-EGFP live attenuated vaccine in controlling viral infections. Owing to the homologous recombination capabilities of CRISPR/Cas9, the Fiber2 gene was subsequently replaced with *egfp*, resulting in the creation of FAdV-4-EGFP-rF2. FAdV-4-EGFP-rF2 is not only highly attenuated in chickens but also provides effective protection against lethal challenges caused by FAdV-4. Furthermore, FAdV-4-EGFP-rF2 induced a comparable level of neutralizing antibodies to that of FA4-EGFP containing Fiber2, suggesting that the genomic location of Fiber2 gene may serve as an insertion site for live attenuated vaccines against FAdV-4 and other pathogens [54].

Using CRISPR/Cas9 technology, Mu et al. [55] successfully constructed the recombinant virus FAdV4-RFP_F1, which expresses RFP and Fiber1 fusion proteins by targeting the N-terminus of Fiber1. This strain is not only highly attenuated in 2-week-old SPF chickens, but also effectively protects against lethal FAdV-4 attacks, making it a promising candidate for a live attenuated vaccine to prevent FAdV-4 disease. Additionally, Lu et al. [56] generated a new recombinant virus, FAdV4-F/8a-rF2, utilizing the CRISPR-Cas9 and Cre-LoxP systems. This strain is highly attenuated and can induce the production of potent neutralizing antibodies in chickens vaccinated with the recombinant virus, providing full protection against FAdV-4 and FAdV-8a. This development lays the groundwork for creating an attenuated bivalent vaccine that addresses the threats posed by two serotypes of FAdVs on farms. An ideal vaccine against FAdVs should target multiple virus serotypes. Therefore, the development of vaccines against more than two serotypes of FAdVs has become a future trend and direction to prevent avian adenovirus infection more efficiently.

Reverse genetic technology has been utilized to modify and attenuate FAdVs strains. The recombinant chimeric virus rR188I, which features a single amino acid substitution at residue 188 of the Hexon protein (where R is mutated to I) of FAdV-4, was developed by Zhang et al. [34] through reverse genetic manipulation. This strain provided full protection against the lethal FAdV-4 challenge, indicating that rR188I is a promising candidate for a live attenuated vaccine. Additionally, Zhang et al. [34] constructed a non-pathogenic chimeric virus rHN20 strain. This rHN20 chimeric virus was generated by replacing the Hexon gene of the highly pathogenic FAdV-4 strain (GenBank No. KU991797), HLJFAd15, with the Hexon gene from the nonpathogenic strain ON1 (GenBank No. GU188428). The immunogenicity of this live vaccine, rHN20, was subsequently evaluated, demonstrating its ability to induce high titers of anti-FAdV-4 neutralizing antibodies and to fully protect immunized chicken flocks from lethal doses of the FAdV-4 virus [57].

Although live attenuated vaccines have demonstrated effective immunity, they face several challenges in practical applications. Issues such as lengthy production times and associated biosafety concerns are notable. Attenuated strains may occasionally undergo virulence reversal or recombination with virulent strains, presenting potential risks to poultry, particularly among susceptible groups. The live FAdV vaccine presents a potential risk of spillover into the environment. Furthermore, some live vaccine strains exhibit significant virulence, raising additional biosafety concerns during the production process and leading to restrictions on their clinical use [58]. Therefore, further foundational research is necessary to support the development of live attenuated vaccines.

**Table 2 ijms-26-06434-t002:** Summary of live attenuated vaccines.

Strain	Origin	Dosage (/Bird)	Inoculation Route	Survival Rate (%)	Reference
rR188I	LMH	10^5^ PFU	IM	100	[34]
FAdV-4/QT35	QT35	5 × 10^4^ TCID_50_	IM	100	[39]
FA4-EGFP	LMH	10^6^ TCID_50_	IM	100	[40]
FAdV4-EGFP-rF2	LMH-F2	2.5 × 10^4^ TCID_50_	IM	100	[54]
FAdV4-RFP-F1	LMH	2 × 10^5^ TCID_50_	IM	100	[55]
FAdV4-F/8a-rF2	LMH	10^6^ TCID_50_	IM	100	[56]
rHN20	LMH	10^6^ PFU	IN	100	[57]
rHN20	LMH	10^6^ PFU	IM	100	[57]
rHN20	LMH	10^6^ PFU	SC	100	[57]

Note: IM, Intramuscular; IN, Intranasal; SC, Subcutaneous.

## 4. Viral Vector Vaccines

The primary viral vector vaccines utilized in poultry include FAdVs, Marek’s disease virus, NDV, herpesvirus of turkey, and fowl poxvirus. Adenovirus-based vectors have emerged as a promising platform for the development of inactivated and live attenuated vaccines [59]. The serotypes of FAdVs primarily utilized for virus vector vaccines development include FAdV-4, FAdV-8, and FAdV-9. Recent advances in poultry DNA recombination technology have significantly enhanced our understanding of the pathogenic mechanisms of fowl viruses and the development of vaccines. Recombinant DNA technology serves as a prevalent strategy for constructing recombinant viruses [60], employing techniques such as the fosmid system [61], CRISPR/Cas9 [55], and Cre-LoxP system [56]. These methodologies are essential tools for manipulating viral genomes, facilitating the precise insertion and deletion of specific sequences. For instance, the recombinant FAdV-9 vector has been demonstrated to effectively express exogenous promoters, leading to significant antibody production and humoral immune responses in the host [62]. This characteristic is essential for developing vaccines that can stimulate strong and specific immune responses against various pathogens. Moreover, the construction of infectious clones from FAdV-4 has enabled direct manipulation of the viral genome, facilitating the rapid generation of adenovirus-based vectors for the expression of specific antigens, such as hemagglutinin of highly pathogenic influenza virus and the VP2 protein of very virulent infectious bursa disease, thereby enhancing the host immune response [51,63,64]. Lu et al. constructed the recombinant FAdV4-F/8a-rF2 strain utilizing the CRISPR-Cas9 and RE-LoXP systems [56]. This virus is not only highly attenuated but also offers complete protection (100%) against challenges posed by FAdV-4 and FAdV-8a. Tian et al. [65] utilized reverse genetic technology to develop a recombinant Newcastle disease virus (NDV) LaSota vaccine strain that expresses the full-length Fiber2 gene of FAdV-4, designated rLaSota-fiber2. A single intramuscular dose of either live or inactivated rLaSota-fiber2 in 2-week-old SPF chickens provided complete protection against NDV challenge. Notably, the intramuscular administration of the live rLaSota-fiber2 vaccine offered superior protection against virulent FAdV-4 challenge and significantly reduced fecal shedding of FAdV-4. These findings suggest that the FAdV-4 vaccine utilizing NDV as a vector represents a promising bivalent vaccine candidate for the control of HHS and NDV, thereby establishing a foundation for the future development of multivaccines targeting multiple serotypes or various viruses. In summary, poultry viral-based vectors represent a substantial advancement in vaccine development, offering a powerful platform for the expression of exogenous genes and the induction of immune responses.

## 5. Subunit Vaccines

The development of subunit vaccines relies on proteins that exhibit strong immunogenicity within the virus. These proteins are expressed in large quantities via genetic engineering techniques and are subsequently combined with adjuvants to immunize animals, thereby eliciting an immune response. The advantages of subunit vaccines include good stability, low cost, and minimal side effects. Additionally, subunit vaccines are considered safer to produce and administer than whole-virus vaccines. Subunit vaccines have been shown to be equally effective as inactivated vaccines while demonstrating good stability and high safety [66]. Importantly, subunit vaccines do not require culture of the entire pathogen, allowing for a more effective and rapid response to outbreaks of FAdVs epidemics.

The immunogenic antigens confirmed in FAdVs primarily include Fiber1, Fiber2, Hexon, and Penton [67]. These four proteins are essential structural components of FAdVs and serve as the principal protective antigens. The relative protection rates following immunization with these different proteins vary, ranging from 40% to 100% [41,68]. Since 2015, significant efforts have been made to develop subunit vaccines to control HHS in poultry flocks. Yin et al. developed a subunit vaccine based on FAdV-4 Fiber2, which elicited robust humoral and cellular immune responses in chickens, providing complete protection against FAdV-4 infection. This development lays a crucial foundation for the effective control of HHS [42]. Schachner et al. investigated the immunogenicity of three recombinant proteins: Fiber1, Fiber2, and Hexon loop-1. In a challenge test, they demonstrated that chickens vaccinated with Fiber2 exhibited significant protection against FAdV-4 infection, thus identifying Fiber2 as an optimal protective antigen for the development of subunit vaccines [68]. Wang et al. [67] expressed Fiber1, Fiber2, Penton, and Hexon of FAdV-4 via the *Escherichia coli* expression system and subsequently compared their immune protective effects on chickens across various vaccination doses. They concluded that Fiber2 demonstrated the most effective protective response and was the preferred candidate for the hydropericardium syndrome (HPS) subunit vaccine, followed by Hexon and Fiber1, whereas Penton provided effective protection only at high doses. The immunogen Fiber2, at a concentration as low as 10 μg/bird combined with Marcol 52 white oil adjuvant, can provide complete protection for SPF chickens [69]. The knob domain serves as the functional region of the viral surface protein Fiber2. Song et al. [70] evaluated the immunogenicity of the recombinant Fiber2-knob protein (F2-knob) as a candidate subunit vaccine against FAdV-4 in 14-day-old SPF chickens. The results showed that antibody levels in 14-day-old SPF chickens immunized with the F2-knob vaccine were significantly higher than in chickens immunized with the inactivated FAdV-4 vaccine. Furthermore, immunization with the F2-knob protein significantly reduced viral shedding and provided complete protection against a virulent FAdV-4 challenge. Additionally, Shah et al. [71] expressed the Penton protein of FAdV-4 via an *E. coli* expression system, subsequently formulating it into a subunit vaccine. Challenge tests confirmed its protective efficacy, suggesting that this subunit vaccine can serve as a candidate agent for the prevention of HPS. Progress has also been made in the research of subunit vaccines for other serotypes. Luca et al. [36] utilized recombinant Fiber with a FAdV-8a genetic background to assess the differences in immune protective effects against homotypic (-8a) and heterotypic (-8b) strains, finally protecting chickens against IBH. The Fib-8a vaccine effectively reduced the viral load in target organs, protected SPF chickens from clinical challenge by homotypic strains, and controlled the adverse effects of isoserotype infection by inducing effective humoral immunity and modulating B-cell and T-cell responses, although it provided no immune protection against the heterotypic FAdV-8b.

Subunit vaccines continue to make significant progress while undergoing continuous improvements. Jia et al. [72] combined the fusion protein DCpe-Fib2 with the surfaces of *Lactococcus lactis* NZ9000 and *Enterococcus faecalis* MDXEF-1 to enhance the Fiber2-specific intestinal mucosal and adaptive immune responses in chickens by targeting dendritic cells (DCs). They evaluated the antiviral effects of these two live recombinant bacteria and reported that chickens orally immunized with *E. faecalis*/pTX8048-DCpep-Fiber2-CWA were completely protected from the FAdV-4 challenge. Recombinant probiotics with Fiber2 bound to their surfaces can stimulate significant humoral and cellular immune responses, thereby alleviating injury to the host post-infection.

Virus-like particles (VLPs) are composed of virus-derived polyprotein structures that can self-assemble. Owing to their structural resemblance to natural viruses in both shape and size, VLPs are capable of effectively eliciting both cellular and humoral immune responses in vivo [73]. Progress has also been made in the development of VLP vaccines for FAdVs. Tufail et al. [74] developed a VLP vaccine by inserting the Hexon protein epitope region of FAdV-4 into the core protein of the hepatitis B virus (HBc). Following immunization of SPF chickens, a robust immune response was elicited, providing up to 90% protection against the FAdV-4 challenge. Additionally, Wang et al. [75] utilized adenovirus vectors to express the capsid proteins Fiber1, Fiber2, and Penton base antigens in cells. These three proteins self-assemble into a penton-dodecahedron (Pt-Dd) antigen. The immune efficacy of this candidate vaccine was evaluated through humoral immune responses, immune molecule expression, and challenge results, revealing that it conferred 100% protection against the FAdV-4 challenge [76]. Despite the numerous advantages of VLP vaccines, the engineering design and large-scale production of VLPs necessitate advanced technology and equipment, which somewhat limits their widespread adoption.

The use of recombinant chimeric proteins has become a popular method for developing subunit vaccines in recent years. Significant progress has also been made in the formulation of recombinant chimeric vaccines on the basis of the Fiber and Hexon antigen epitopes of FAdV-4. Aziz et al. [77] assessed the immune protection efficacy of both full-length Penton base and truncated Penton base (1–255 aa), revealing that both vaccines achieved a protection rate of only 50%. In another study, Hu et al. utilized a prokaryotic expression system to develop a fusion subunit antigen (rFH), which comprises a truncated fragment of Fiber2 (Gly 275-Pro 479 aa) and the epitope coding sequence of Hexon (Met 21-Val 51 aa). The results from viral challenge trials demonstrated that high doses (≥5 μg/bird) of the rFH vaccine could confer complete protection for chickens [78]. The Hess team focused on studying cross-serotype chimeric proteins by exchanging Fiber head fragments between FAdV-8a and FAdV-8b to create new chimeras, crecFib-8a/8b and crecFib-8b/8a. Compared with single Fiber, crecFib chimeras elicited cross-neutralizing antibodies that simultaneously protected chickens from challenges with either FAdV-8a or FAdV-8b, significantly reducing viral loads in target organs [79]. Furthermore, the Hess team designed a new chimera, crecFib-4/11, which combines the epitopes of two different species, FAdV-4 and FAdV-11. This chimeric protein has been shown to be an effective protective strategy capable of safeguarding chickens from both HHS and IBH and is anticipated to provide cross-serotype protection against FAdVs [80]. The details of these subunit vaccines are summarized in Table 3.

## 6. Vaccine Design and Prospects

FAdVs infection is prevalent worldwide, with numerous serotypes exhibiting significant variability in pathogenicity. Although various types of vaccines have been developed for virulent FAdV-4 strains and have demonstrated good immune efficacy in controlling this disease, the existence of multiple FAdVs serotypes and the absence of cross-protection between different serotypes pose significant challenges. Currently, multiple serotypes of FAdVs remain prevalent within the same area or same poultry farm [32]. Co-infection with immunosuppressive diseases in poultry, such as the infectious bursal disease virus (IBDV), reticuloendothelial hyperplasia virus, avian leukemia virus, and chicken infectious anemia virus, can impair the host immune system and enhance the pathogenicity of certain FAdVs. Additionally, non-infectious factors, including nutritional deficiencies, improper medication, exposure to mycotoxins, and environmental or management-related stress, can place poultry flocks in a suboptimal health state, thereby increasing the prevalence and severity of FAdV infections [9,33,83]. Therefore, future research should prioritize the development of multivaccines that not only target FAdVs diseases but also provide immune protection against other poultry diseases.

More recently, the development of live viral vectors and delivery systems for antigens has gained prominence [84]. The development of vaccine carriers and adjuvants utilizing nanomaterials represents a novel research direction and trend in the future advancement of animal vaccines [85,86]. This approach offers multifaceted solutions and introduces innovative scientific concepts for the effective prevention of FAdVs. Moreover, vaccines formulated with a new generation of adjuvants and delivery systems have the potential to not only generate high-titer antibodies but also stimulate CD4+ T-cell and/or CD8+ T-cell responses from the host immune system [66]. Additionally, nanoparticles can encapsulate viral antigens or display them on their surfaces, effectively mimicking the structure of viruses. This strategy enhances the immune system’s recognition capabilities and could induce robust humoral and cellular immune responses in the host. Furthermore, these vaccines can significantly reduce viral shedding and alleviate clinical symptoms of the disease following exposure to virulent strains. Xu et al. [87] developed a manganese-silicon nanoplatform, MnOx@HMSN (CDA), which increased the activation of the STING pathway and increased the expression of type I interferons and other pro-inflammatory cytokines in dendritic cells. MnOx@HMSN, loaded with the SARS-CoV-2 antigen, achieved a strong and long-lasting (up to one year) humoral immune response and exhibited neutralizing capacity in BALB/c mice. These findings indicate that MnOx@HMSN (CDA) represents a multifunctional nanoplatform for vaccine development. Hills et al. [88] linked the receptor-binding domain tandem peptide sequence of a SARS-like β-coronavirus to a protein nanocage, resulting in the creation of a quadruple nanocage that can elicit a broad antiviral response following immunization. Zhou et al. [89] loaded the head region of the soluble glycoprotein G of Nipah virus (NiV-sG) as an antigen onto ferritin-based self-assembled nanoparticles via spycatcher/spytag technology, resulting in NiV-g-ferritin nanoparticle vaccines. Notably, the NiV-g-ferritin vaccine elicited faster, broader, and higher serum neutralization responses than did the NiV-sG vaccine. These research studies offer novel insights into the development of FAdVs vaccines based on ferritin nanoparticle carriers. The complementary and ongoing exploration of genetic engineering, adjuvants, and nano-delivery systems may lead to the development of more effective vaccines that provide robust protection against avian viral diseases, thereby contributing to healthy and sustainable advancements in the poultry industry and global food security.

Inducing robust cellular immunity through vaccine immunization, particularly through antigens targeted at DCs that elicit T-cell responses [90], is a crucial strategy for developing safer and more effective FAdVs vaccines. Researchers are continually exploring novel production methods, improving vaccine formulations, and broadening the scope of applications to address future FAdVs epidemic challenges more effectively.

## 7. Conclusions

FAdVs have emerged as a significant epidemic threat worldwide. The existence of multiple FAdVs serotypes, combined with their extensive host range and capacity for both horizontal and vertical transmission, underscores their status as critical pathogens in the poultry industry, leading to substantial economic losses on a global scale. This review focuses on vaccines developed and studied worldwide for various serotypes of FAdVs in the past decade. It encompasses inactivated vaccines, live attenuated vaccines (including host-adapted attenuated vaccines, gene deletion vaccines), viral vector vaccines, and subunit vaccines (including VLP proteins and chimeric proteins). The current limitations and future development directions of FAdVs vaccine development are also proposed to provide a reference for new-generation vaccines and innovative vaccination strategies against FAdVs, as well as for the rapid development of highly effective vaccines.

## Figures and Tables

**Figure 1 ijms-26-06434-f001:**
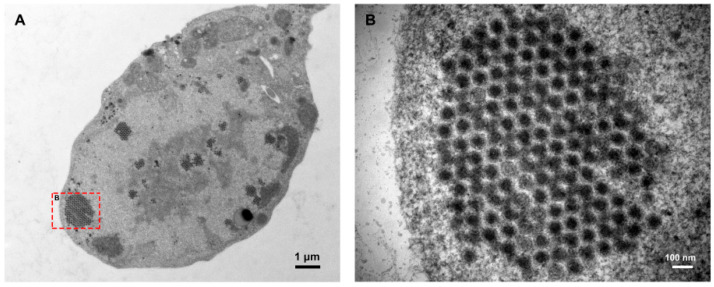
The virus particles of FAdV-4 are arranged in a crystal lattice pattern in the nucleus of liver. Panel (**B**) shows local magnified views of panel (**A**), indicated by the red dashed box.

**Figure 2 ijms-26-06434-f002:**
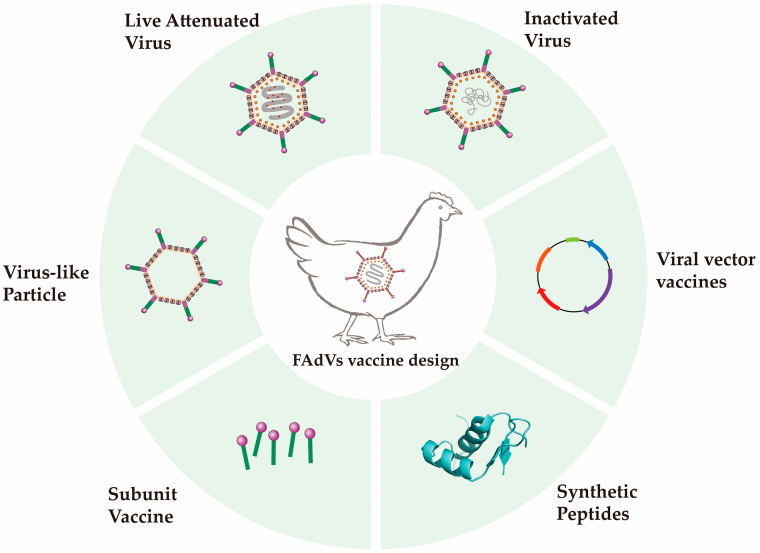
The FAdVs vaccine design strategy.

**Table 1 ijms-26-06434-t001:** Summary of inactivated vaccines against FAdVs.

Strain	Origin	Adjuvant	Dosage (/Bird)	Inoculation Route	Survival Rate (%)	Reference
K531/07	CEL	ISA 70	5 × 10^5^ TCID_50_	IM	80	[38]
SB15	LMH	Oil	1 × 10^6^ TCID_50_	SC	100	[41]
HLJFAd15	CEL	Oil	1 × 10^6^ TCID_50_	IM	100	[45]
SDJN0105	CEL	Oil	1 × 10^6^ TCID_50_	IM	100	[46]
CH/GZXF/1602	CEK	Oil	1 × 10^6^ TCID_50_	SC	100	[47]
FA4-F8b	LMH	Oil	1 × 10^6^ TCID_50_	IM	100	[48]
rFAdV-4-fiber/8b	LMH	ADJ501	1 × 10^6^ TCID_50_	IM	100	[49]
FAdV-4 rHN20-vvIBDV-VP2	LMH	Oil	3 × 10^6^ PFU	IM	100	[51]

Note: CEL, Chicken embryo liver cells; CEK, Chicken embryo kidney cells; LMH, Leghorn male hepatocellular; IM, Intramuscular; SC, Subcutaneous.

**Table 3 ijms-26-06434-t003:** Summary of subunit vaccines.

Proteins	Adjuvant	Dose	Inoculation Route	Survival Rate (%)	Reference
rFiber2	FCA	50 μg	IM	80	[42]
rFiber2	FCA	100 μg	IM	100	[42]
rFiber2	FCA	150 μg	IM	100	[42]
Fiber2	FCA	2.5 μg	SC	100	[41]
Fiber1	FCA	100 μg	ND	100	[67]
Fiber2	FCA	50 μg	ND	100	[67]
Penton	FCA	200 μg	ND	90	[67]
Hexon	FCA	200 μg	ND	95	[67]
Fiber2	Marcol 52 white oil	10 μg	SC	100	[69]
Fiber2	Marcol 52 white oil	5 μg	SC	90	[69]
Penton	Marcol 52 white oil	200 μg	SC	100	[69]
Penton	Marcol 52 white oil	100 μg	SC	70	[69]
Penton base	FCA	200 μL	SC	90	[71]
His_6_-PreS-penton base^1−225^	Montanide ISA71 VG	500 μL	SC	50	[77]
His_6_-PreS-penton base FL	Montanide ISA71 VG	500 μL	SC	50	[77]
*Enterococcus faecalis*/ pTX8048-DCpep-Fiber2-CWA	N/A	5.0 × 1.0 × 10^9^ CFU	OG	100	[72]
Pt-Dds	N/A	0.5 μg	ND	100	[75]
rFH	Montanide^TM^ ISA71 VG	≥5 μg	IM	100	[78]
HBc-hexon(Asp348-Phe369)	Montanide^TM^ ISA71 VG	100 μg	SC	90	[74]
HBc-hexon (Ser19-Pro82)	Montanide^TM^ ISA71 VG	100 μg	SC	70	[74]
HBc-hexon(Gly932-Phe956)	Montanide^TM^ ISA71 VG	100 μg	SC	40	[74]
Fib-8a	GERBU Adjuvant P	50 μg	IM	94	[36]
Fiber1/2 knob domain	FCA	10 μg	IM	100	[81]
Fiber2	FCA	10 μg	IM	100	[81]
FliBc-fiber2-SP	white oil	50 μg	IM	100	[82]

Note: FCA, Freund’s complete adjuvant; IM, Intramuscular; SC, Subcutaneous; ND, Not noted; OG, orally gavage.

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
