# Peer review of "Advances and Prospects of Fowl Adenoviruses Vaccine Technologies in the Past Decade"

_ijms, 2025, doi:10.3390/ijms26136434_

Round 1
Reviewer 1 Report
Comments and Suggestions for Authors
The review by Zhu et al. provides a thorough summary of developed vaccines against fowl adenoviruses. However, it could be improved in several areas.
Most importantly, the authors frequently refer to Group I fowl adenoviruses, which is a significant misunderstanding as there has never been such a grouping. The genus Aviadenovirus used to be divided into three groups, but this taxonomy has been discontinued and rearranged more than 20 years ago. Group II (duck adenovirus 1, also known as egg drop syndrome virus) was moved to the genus Barthadenovirus (formerly Atadenovirus), and Group III (turkey adenovirus 3, also known as turkey haemorrhagic enteritis virus) was moved to the genus Siadenovirus. All viruses named as "fowl adenovirus" (serotypes fowl adenovirus 1–8a and 8b–11) remain in the genus Aviadenovirus. Please remove and correct all instances of this error.
Latest taxonomic reference:
Full text:
https://ictv. global/report/chapter/adenoviridae/Adenoviridae
Citation:
Benkő M, Aoki K, Arnberg N, Davison AJ, Echavarría M, Hess M, Jones MS, Kaján GL, Kajon AE, Mittal SK, Podgorski II, San Martín C, Wadell G, Watanabe H, Harrach B, Ictv Report Consortium. ICTV Virus Taxonomy Profile: Adenoviridae 2022. J Gen Virol. 2022 Mar;103(3):001721. doi: 10. 1099/jgv. 0. 001721. PMID: 35262477; PMCID: PMC 9176265.
Another concern is that while the authors provide an informative summary figure (Fig. 2) of the various vaccine technologies employed, they do not discuss some of these (E.g., DNA vaccines: PMID: 34649009), or others are presented in a disorganised manner within one subchapter. For example, vector vaccines might be attenuated through gene knock- outs, yet it seems inappropriate to include these vaccines in the Live attenuated vaccines subchapter. Moreover, the vector constructs developed by Eva Nagy are not referenced (PMID: 29495283, 29269248, 28374242, 33915232, 28780115). In summary, a separate discussion on vector vaccines is recommended, while a discussion on virus- like particles may also be valuable.
Lastly, additional recommended edits, comments, and questions are outlined in the attached, downloadable commented PDF file; please include these in your response as well.

Author Response
Reviewer 1:
Most importantly, the authors frequently refer to Group I fowl adenoviruses, which is a significant misunderstanding as there has never been such a grouping. The genus Aviadenovirus used to be divided into three groups, but this taxonomy has been discontinued and rearranged more than 20 years ago. Group II (duck adenovirus 1, also known as egg drop syndrome virus) was moved to the genus Barthadenovirus (formerly Atadenovirus), and Group III (turkey adenovirus 3, also known as turkey haemorrhagic enteritis virus) was moved to the genus Siadenovirus. All viruses named as "fowl adenovirus" (serotypes fowl adenovirus 1–8a and 8b–11) remain in the genus Aviadenovirus. Please remove and correct all instances of this error.
Latest taxonomic reference:
Full text:
https://ictv.global/report/chapter/adenoviridae/Adenoviridae
Citation:
Benkő M, Aoki K, Arnberg N, Davison AJ, Echavarría M, Hess M, Jones MS, Kaján GL, Kajon AE, Mittal SK, Podgorski II, San Martín C, Wadell G, Watanabe H, Harrach B, Ictv Report Consortium. ICTV Virus Taxonomy Profile: Adenoviridae 2022. J Gen Virol. 2022 Mar;103(3):001721. doi: 10. 1099/jgv. 0. 001721. PMID: 35262477; PMCID: PMC 9176265.
Another concern is that while the authors provide an informative summary figure (Fig. 2) of the various vaccine technologies employed, they do not discuss some of these (E.g., DNA vaccines: PMID: 34649009), or others are presented in a disorganised manner within one subchapter. For example, vector vaccines might be attenuated through gene knock- outs, yet it seems inappropriate to include these vaccines in the Live attenuated vaccines subchapter. Moreover, the vector constructs developed by Eva Nagy are not referenced (PMID: 29495283, 29269248, 28374242, 33915232, 28780115). In summary, a separate discussion on vector vaccines is recommended, while a discussion on virus- like particles may also be valuable.
Thank you for your constructive suggestions. We have adjusted the relevant information in Figure 2 to enable it to more intuitively and accurately convey the Fowl Adenovirus (FAdVs) vaccine information pertinent to this manuscript. Additionally, we supplemented the related references from Eva Nagy and established a separate discussion on Fowl Adenovirus-based vectors. However, we found it challenging to discuss this topic independently, as vaccines based on Fowl Adenovirus vectors encompass both inactivated and live attenuated vaccines. Furthermore, we also addressed virus-like particle vaccines; however, it was not feasible to separate them independently, given that virus-like particle vaccines are themselves a type of subunit vaccine.
Lastly, additional recommended edits, comments, and questions are outlined in the attached, downloadable commented PDF file; please include these in your response as well.
- Line 2, Line9, Line21 “group I”, and Line9 “subgroup” outdated, incorrect terminology
Done as requested. We corrected these words.
- Line10 and Line32 “Adenoviridae”, Viral taxon names are italicised. Line322 “Lactococcus lactis” should be italics
Done as requested.
- Line32 “categorized into three subgroups: I, II, and III.” This is not and has never been true, it is a huge misunderstanding. The genus Aviadenovirus used to be divided into 3 groups, but this taxonomy has been discontinued and rearranged for more than 20 years. Group II (duck adenovirus 1 = egg drop syndrome virus) was moved to genus Bartadenovirus (formerly Atadenovirus), and Group III (turkey adenovirus 3 = turkey hemorrhagic enteritis virus) was moved to genus Siadenovirus. All viruses named as "fowl adenovirus" (the serotypes fowl adenovirus 1-8a and 8b-11) remained in the genus Aviadenovirus.
Please remove, correct all instances of this error.
Latest taxonomic reference:
Full text:
https://ictv.global/report/chapter/adenoviridae/adenoviridae
Citation:
Benkő M, Aoki K, Arnberg N, Davison AJ, Echavarría M, Hess M, Jones MS, Kaján GL, Kajon AE, Mittal SK, Podgorski II, San Martín C, Wadell G, Watanabe H, Harrach B, Ictv Report Consortium. ICTV Virus Taxonomy Profile: Adenoviridae 2022. J Gen Virol. 2022 Mar;103(3):001721. doi: 10.1099/jgv.0.001721. PMID: 35262477; PMCID: PMC9176265.
We meticulously reviewed the ICTV category and implemented rigorous modifications to the relevant sections of the manuscript. For further details, please refer to our manuscript. Thanks again for your constructive suggestions.
- Line35 “group I”, Line 48 “and subgenus”, Line92 “and hydropericardium syndrome (HPS)”, Line134 “New genotypes of”, Line243 “immune”, Line439 “Group I” should be deleted
Done as requested.
- Line 36 “FAdV-A to FAdV-E” Again, outdated species name. Current binomial species names:
Aviadenovirus ventriculi (formerly Fowl A)
- quintum (Fowl B)
- hydropericardii (Fowl C)
- gallinae (Fowl D)
- hepatitidis (Fowl E)
Please correct all instances in the MS.
Reference: see above
Thanks for your constructive suggestions. Done as requested.
- Line 43 “Penton” should be “Penton base”
Done as requested.
- Line52 “FAdV‐4”, Line81 “FAdVs”, Line94 “FAdVs” change to “FAdV”
Done as requested.
- Line83 “genotype” The word "strain" would be recommended. The new variant is not that distant from the reference strain of FAdV-4, it definitely does not represent a novel type.
Done as requested. We corrected it to novel variant strain
- Line94 “our country” changes to “China”
Done as requested.
- Table 1 “Survival rate” After a trial infection with the FAdV strain used in the vaccine?
Yes, it’s the survival rates of chickens post-FAdV-infection used in the vaccine administration.
- Line187 Vector vaccines are usually developed from apathogenic (and not attenuated) strains for obvious safety reasons. Why are they discussed in this subchapter?
Thanks for your suggestions. We acknowledge that our expression in the manuscript may have caused confusion; therefore, we have decided to remove this subchapter.
- Line190 reference 52, Instead of a Newcastle disease virus vector expressing IBV proteins, the authors could cite at least some of the numerous papers by Eva Nagy. E.g.: PMID: 29495283, 29269248, 28374242, 33915232, 28780115
Thanks for your suggestions. We have supplemented the discussion on fowl adenovirus-based vectors and cited the relevant references accordingly.
- Line194 reference 53. This reference is about nanoparticle vaccines. It is neither about attenuated nor about vector vaccines. Why is it used here? Please cite relevant references.
Done as requested. We have cited the relevant references accordingly.
- Line197 This paragraph seems to be a mixture of unrelated sentence groups. First, the authors start with traditional live attenuated vaccines (197-202). Then turn to recombinant, genome-edited vaccines. This Reviewer understands that these novel vaccines are also attenuated and they are called live-attenuated by the original authors. However, a more distinguishing approach seems more logical to this Reviewer: where traditional, serial passaged and attenuated vaccine strains are discussed separated from the genome-edited, artificially modified (GMO) recombinant or vector vaccine strains.
We have reorganized the language and presented the relevant content in the manuscript. We acknowledge your suggestion that discussing traditional, serial passaged, and attenuated vaccine strains separately may be more beneficial. However, due to the limited number of reports concerning live attenuated vaccines for avian adenoviruses, we have opted to consolidate the relevant content. Thank you once again for your suggestions.
- Line215 “cells expressing Fiber-2” Very easy to misunderstand that the virus strain expresses fiber 2 and not the special cells. Pls reformulate and stress, that special fiber-2-expressing cells are needed.
Thanks for your suggestion. Done as requested.
- Line218 “Fiber-2” changes to “the genomic location of fiber 2 gene”
Done as requested.
- Line235 “rR188I” What is modified in this strain's genome?
Here it means rR188I mutant in the hexon protein. A Single Amino Acid at Residue 188 (R was mutated into I) of the Hexon Protein Is Responsible for the Pathogenicity of the Emerging Novel Virus Fowl Adenovirus 4.
- Line236 “can be neutralized by serum both in vitro and in vivo” What is meant here?
The rR188I mutant strain was significantly neutralized by chicken serum in vitro and in vivo, whereas the wild-type strain was able to replicate efficiently. Maybe our expression made you confused, so we corrected it in the manuscript.
- Line362 “serotypes” changes to “species” because FAdV-8a and 8b are different serotypes already (l. 356-361). FAdV-4 and -11 are already from two different species.
Done as requested.

Reviewer 2 Report
Comments and Suggestions for Authors
Dear authors
The following recommendations should be considered to be included in this review article, particularly recommending the edition in the Introduction section, addition of a Discussion section and improving the content and length of the Conclusion section as well.
-. You are are aware about both horizontal and vertical transmissions of FAdVs in chicken operations from breeders to broilers (L74-L76). You are recommended to assume the role and importance of the immunization at the broiler breeder flock and/or broiler chicken flock levels to clarify the application and benefits of FAdV candidates for vaccination.
-. You included the main FAdV-syndrome currently hitting the worldwide chicken industry (L77-L82). Please briefly include the main gross and microscopic features, morbi and mortality rates and economic losses of each of these FAdV-related diseases
-. The authors mentioned in L387-L388, "This approach offers multifaceted solutions and introduces innovative scientific concepts for the effective prevention and control of FAdVs". This brief sentence could be an introductory concept not only for the study and development of vaccine candidates against multiple and pathogenic serotypes/genotypes of field FAdV isolates, but also to consider that immunization is one of the necessary tools to be implemented here. Please include relevant preventive field measures and laboratory diagnostic testing to be consider on this topic.
-. The authors mentioned the synergistic pathologic action between pathogenic FAdV, infectious bursal disease virus and chicken infectious anemia virus in the field (L15, L104 and L376). This is correct, but other multiple factors can interact with pathogenic FAdVs under field conditions and lead the immunosuppressive negative impact in broiler progeny mainly. Please list the wide range of non-infectious and infectious entities potentially causing suboptimal conditions of chicken flocks, which facilitate the pathogenic and immunosuppressive actions of FAdV field challenge isolates.
Author Response
Dear authors,
The following recommendations should be considered to be included in this review article, particularly recommending the edition in the Introduction section, addition of a Discussion section and improving the content and length of the Conclusion section as well.
We thank the reviewers for your constructive comments. To address the concerns raised we have rigorous modifications to the relevant sections of the manuscript according to your comments. And we used yellow highlighting to show the changes we have made in the revised manuscript. Below is a detailed point-by-point response to the concerns raised.
- You are aware about both horizontal and vertical transmissions of FAdVs in chicken operations from breeders to broilers (L74-L76). You are recommended to assume the role and importance of the immunization at the broiler breeder flock and/or broiler chicken flock levels to clarify the application and benefits of FAdV candidates for vaccination.
Thanks for your suggestion. Done as requested.
- You included the main FAdV-syndrome currently hitting the worldwide chicken industry (L77-L82). Please briefly include the main gross and microscopic features, morbi and mortality rates and economic losses of each of these FAdV-related diseases.
FAdVs infection can lead to various clinical syndromes. Notably, FAdV-1 is associated with adenoviral gizzard erosion, while FAdV-4 is linked to hepatitis-hydropericardium syndrome (HHS). Additionally, FAdV-2, FAdV-11, FAdV-8a, and FAdV-8b are known to cause inclusion body hepatitis (IBH). Among these, FAdV-4 poses the greatest threat to the poultry industry. The impact of other serotypes is relatively minor, with only sporadic reports regarding the development of related vaccines. Clinical cases of FAdVs infection have been extensively documented in poultry populations worldwide, however, the clinical symptoms are often atypical. Therefore, this manuscript focuses on the clinical symptoms, morbidity, and mortality rates associated with FAdV-4, while providing limited discussion on other serotypes.
- The authors mentioned in L387-L388, "This approach offers multifaceted solutions and introduces innovative scientific concepts for the effective prevention and control of FAdVs". This brief sentence could be an introductory concept not only for the study and development of vaccine candidates against multiple and pathogenic serotypes/genotypes of field FAdV isolates, but also to consider that immunization is one of the necessary tools to be implemented here. Please include relevant preventive field measures and laboratory diagnostic testing to be consider on this topic.
Thanks for your suggestion. This topic focuses on the research of fowl adenoviruses vaccines; therefore, we have not included relevant content such as preventive field measures and laboratory diagnostic testing. Additionally, there are already existing reports concerning preventive measures and laboratory diagnoses. Reference:Fowl adenovirus serotype 4: Epidemiology, pathogenesis, diagnostic detection, and vaccine strategies. Poult Sci. 2017 Aug 1;96(8):2630-2640. doi: 10.3382/ps/pex087.
- The authors mentioned the synergistic pathologic action between pathogenic FAdV, infectious bursal disease virus and chicken infectious anemia virus in the field (L15, L104 and L376). This is correct, but other multiple factors can interact with pathogenic FAdVs under field conditions and lead the immunosuppressive negative impact in broiler progeny mainly. Please list the wide range of non-infectious and infectious entities potentially causing suboptimal conditions of chicken flocks, which facilitate the pathogenic and immunosuppressive actions of FAdV field challenge isolates.
Thanks for your suggestions. We have listed the related contents in the manuscript. Please see details in the Line 382-388.

Round 2
Reviewer 1 Report
Comments and Suggestions for Authors
The review by Zhu et al. was enhanced significantly from a taxonomical point of view. However, this Reviewer is still not satisfied with the main structure of the review, that vector vaccines are discussed under the title “Live attenuated vaccines” for example. If the Authors and the Editor can accept this structure, this Reviewer will not mention it again.
Please use Word's “Track changes” feature in the future, not the yellow background. This way, reviewers see deletions as well as additions.
Concerning the question about the mutant vaccine strain rR188I: What is modified in this strain's genome? Thank you for answering this question. This Reviewer would like to make a very general comment, sorry for being so stubborn. When a reviewer asks a question, he/she does not simply require an answer. The reviewer would like to see the answer edited into the manuscript. The reviewer believes that this might interest other readers as well; and that this part requires further clarification.
Comments on the Quality of English Languagel. 243-244: The new, shortened sentence is grammatically incorrect. There is no predicate in this sentence.
Author Response
1. The review by Zhu et al. was enhanced significantly from a taxonomical point of view. However, this Reviewer is still not satisfied with the main structure of the review, that vector vaccines are discussed under the title “Live attenuated vaccines” for example. If the Authors and the Editor can accept this structure, this Reviewer will not mention it again.
We thank the reviewers for your constructive comments. To address the concerns raised we have rigorous modifications to the relevant sections of the manuscript according to your comments. The Adenovirus-based vectors vaccines are discussed as a separate section.
And we used “Track changes” to show the changes we have made in the revised manuscript. Below is a detailed point-by-point response to the concerns raised.
2. Please use Word's “Track changes” feature in the future, not the yellow background. This way, reviewers see deletions as well as additions.
Thanks for your suggestions. Done as requested.
3. Concerning the question about the mutant vaccine strain rR188I: What is modified in this strain's genome? Thank you for answering this question. This Reviewer would like to make a very general comment, sorry for being so stubborn. When a reviewer asks a question, he/she does not simply require an answer. The reviewer would like to see the answer edited into the manuscript. The reviewer believes that this might interest other readers as well; and that this part requires further clarification.
Done as requested. Here it means rR188I mutant in the hexon protein. A single amino acid at residue 188 (R was mutated into I) of the Hexon protein is responsible for the pathogenicity of the emerging novel virus fowl adenovirus 4. We supplemented the related information in the manuscript.
4. Line 243-244: The new, shortened sentence is grammatically incorrect. There is no predicate in this sentence.
Thanks for your suggestions. The sentence has been revised. Please see lines 228–229 for details.

Reviewer 2 Report
Comments and Suggestions for Authors
Dear authors
You addressed all comments and suggestions previously made. I have only one minor edit still to suggest in L384, in which "improper drug use" could be modified.
Author Response
You addressed all comments and suggestions previously made. I have only one minor edit still to suggest in L384, in which "improper drug use" could be modified.
Thanks for your suggestion. We changed "improper drug use" to "improper medication" (Line 398).